# Functional Indirection Neural Estimator for Better Out-of-distribution Generalization

**Kha Pham**[1]    **Hung Le**[1]    **Man Ngo**[2]    **Truyen Tran**[1]
[1] Applied Artificial Intelligence Institute, Deakin University
[2] Faculty of Mathematics and Computer Science, VNUHCM-University of Science

[1] {phti, thai.le, truyen.tran}@deakin.edu.au
[2] nmman@hcmus.edu.vn

## Abstract

The capacity to achieve out-of-distribution (OOD) generalization is a hallmark of human intelligence and yet remains out of reach for machines. This remarkable capability has been attributed to our abilities to make conceptual abstraction and analogy, and to a mechanism known as indirection, which binds two representations and uses one representation to refer to the other. Inspired by these mechanisms, we hypothesize that OOD generalization may be achieved by performing analogy-making and indirection in the *functional* space instead of the data space as in current methods. To realize this, we design FINE (Functional Indirection Neural Estimator), a neural framework that *learns to compose functions* that map data input to output on-the-fly. FINE consists of a backbone network and a trainable semantic memory of basis weight matrices. Upon seeing a new input-output data pair, FINE dynamically constructs the backbone weights by mixing the basis weights. The mixing coefficients are indirectly computed through querying a separate corresponding semantic memory using the data pair. We demonstrate empirically that FINE can strongly improve out-of-distribution generalization on IQ tasks that involve geometric transformations. In particular, we train FINE and competing models on IQ tasks using images from the MNIST, Omniglot and CIFAR100 datasets and test on tasks with unseen image classes from one or different datasets and unseen transformation rules. FINE not only achieves the best performance on all tasks but also is able to adapt to small-scale data scenarios.

## 1 Introduction

*Every computer science problem can be solved with a higher level of indirection.*

—Andrew Koenig, Butler Lampson, David J. Wheeler

Generalizing to new circumstances is a hallmark of intelligence [16, 4, 11]. In some Intelligence Quotient (IQ) tests–a popular benchmark for human intelligence–one must leverage their prior experience to identify the hidden abstract rules out of a concrete example (e.g., a transformation of an image) and then apply the rules to the next (e.g., a new set of images of totally different appearance). These tasks necessitate several key capabilities, including conceptual *abstraction* and *analogy-making* [22]. Abstraction allows us to extend a concept to novel situations. It is also driven by analogy-making, which maps the current situation to the previous experience stored in the memory. Indeed, analogy-making has been argued to be one of the most important abilities of human cognition, or even further, "a concept is a package of analogies" [12]. The ability for humans to traverse seamlessly across concrete and abstraction levels suggests another mechanism known as *indirection* to bind two representations and use one representation to refer to the other [16, 20].

36th Conference on Neural Information Processing Systems (NeurIPS 2022).

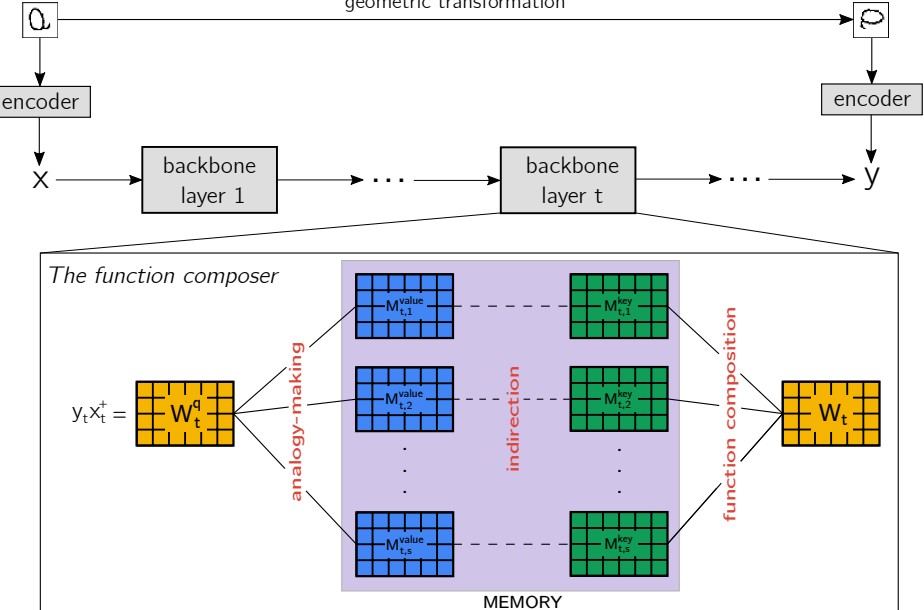

Figure 1: FINE architecture. *Above:* FINE uses a pre-defined deep backbone architecture to approximate a function mapping a given input embedding $x$ to a given output embedding $y$. *Below:* given the input $x_t$ and pseudo-output $y_t$ of the $t^{\text{th}}$ backbone layer, FINE first computes the query $W_t^q$ which represents the relation between $x_t$ and $y_t$. Then FINE performs analogy-making to compare the query with past experiences in the form of value memories. Finally, FINE binds the value memories with associated key memories via indirection and computes the weight $W_t$ for the $t^{\text{th}}$ backbone layer.

Several deep learning models have successfully utilized analogy and indirection. The Transformer [30] and RelationNet [28] learn analogies between data, through self-attention or pairwise functions. The ESBN [33] goes further by incorporating the indirection mechanism to bind an entity to a symbol and then reason on the symbols; and this has proved to be efficient on tasks involving abstract rules, similar to those aforementioned IQ tests. However, a common drawback of these approaches is that they operate on the data space, and thus are susceptible to out-of-distribution samples.

In this paper, we propose to perform analogy-making and indirection in *functional* spaces instead. We aim to *learn to compose* a functional mapping from a given input to an output *on-the-fly*. By doing so, we achieve two clear advantages. First, since the class of possible mappings is often restricted, it may not require a large amount of training data to learn the distribution of functions. Second, more importantly, since this approach performs indirection in functional spaces, it avoids bindings between numerous entities and symbols in data spaces, thus may help improve the out-of-distribution generalization capability.

To this end, we introduce a new class of problems that requires functional analogy-making and indirection, which are deemed to be challenging for current data-oriented approaches. The tasks are similar to popular IQ tasks in which the model is given hints about the hidden rules, then it has to predict the missing answer following the rules. One reasonable approach is that models should be able to compare the current task to what they saw previously to identify the rules between appearing entities, and thus has to search on functional spaces instead of data spaces. More concretely, we construct the IQ tasks by applying geometric transformations to images from MNIST dataset, hand-written Omniglot dataset and real-image CIFAR100 dataset, where the training set and test set contain disjoint image classes from the same or different datasets, and possibly disjoint transformation rules .

Second, we present a novel framework named Functional Indirection Neural Estimator (FINE) to solve this class of problems (see Fig. 1 for the overall architecture of FINE). FINE consists of (a) a neural backbone to approximate the functions and (b) a trainable key-value memory module that stores the basis of the network weights that spans the space of possible functions defined by the backbone. The weight basis memories allow FINE to perform analogy-making and indirection in the function space. More concretely, when a new IQ task arrives, FINE first (1) takes the hint images to make analogies with value memories, then (2) performs indirection to bind value memories with key memories and finally (3) computes the approximated functions based on key memories. Throughout

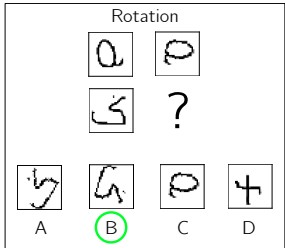 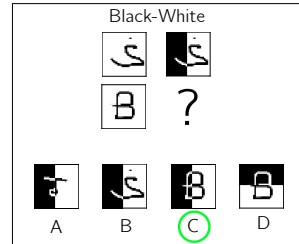

Figure 2: Examples of two IQ tasks involving geometric transformations. Choices with green circles are the correct solutions. *Left.* 90-degree rotation. *Right.* syntactic black-white transformation: a part of the image is transformed to the opposite colors.

a comprehensive suite of experiments, we demonstrate that FINE outperforms the competing methods and adapts more effectively in small data scenarios.

## 2 Tasks

For concreteness, we will focus on Intelligence Quotient (IQ) tests, which have been widely accepted as one of the reliable benchmarks to measure human intelligence [26]. We will study a popular class of IQ tasks that provides hints following some hidden rules and requires the player to choose among given choices to fill in a placeholder so that the filled-in entity obeys the rules of the current task. In order to succeed in these tasks, the player must be able to figure out the hidden rules and perform analogy-making to select the correct choice. Moreover, once figuring out the rules for the current task, a human player can almost always solve tasks with similar rules regardless of the appearing entities given in the tasks. This remarkable ability of out-of-distribution generalization indicates that humans treat objects and relations abstractly instead of relying on the raw sensory data.

We aim to solve IQ tasks that involve geometric transformations (e.g., see Fig. 2 for an example), which include affine transformations (translation, rotation, shear, scale, and reflection), non-linear transformations (fisheye, horizontal wave) and syntactic transformations (black-white, swap). Details of transformations are given in Supplementary. In a task, the models are given 3 images $x, y$ and $x'$, where $y$ is the result of $x$ after applying a geometric transformation. The models are then asked to select $y'$ among 4 choices $y_1, y_2, y_3, y_4$ so that $(x', y')$ follows the same rule as $(x, y)$ (i.e. if $y = f(x)$ then $y' = f(x')$ for transformation $f$). The 4 choices include (i) one with correct object/image and correct transformation (which is the solution), (ii) one with correct object/image and incorrect transformation, (iii) one with incorrect object/image and correct transformation, and (iv) one with both incorrect object/image and transformation.

Inspired by human capability, a reasonable approach to solve these tasks is that models should be able to figure out the transformation (or relation) between objects/images and apply the transformation to novel objects/images. The datasets can be classified into two main categories: single-transformation datasets and multi-transformation datasets. Single-transformation datasets are ones that only include a particular transformation, e.g. rotation. Note that the individual transformations of the same type vary, e.g., rotations by different angles. Multi-transformation datasets, on the other hand, consist of several transformation types. To test the generalization capability of the models, we build testing sets including classes of images that have never been seen during training (see Section 4.1 and Section 4.2), or even more challenging tasks including unseen rules and unseen datasets (see Section 4.3). Models must be able to leverage knowledge and memory gained from the training dataset to solve a new task.

## 3 Method

### 3.1 Functional Hypothesis

Let $\mathcal{X}$ and $\mathcal{Y}$ be the data input and output spaces, respectively. Denote by $(\mathcal{X}_{\text{train}}, \mathcal{Y}_{\text{train}}) \subset (\mathcal{X}, \mathcal{Y})$ the training set, and $(\mathcal{X}_{\text{test}}, \mathcal{Y}_{\text{test}}) \subset (\mathcal{X}, \mathcal{Y})$ the non-overlapping test set. Classical ML assumes that $\mathcal{X}_{\text{train}}$ and $\mathcal{X}_{\text{test}}$ are drawn from the same distribution. Under this hypothesis, it is reasonable (for frequentists) to find a function $f : \mathcal{X} \to \mathcal{Y}$ in the functional space $\mathcal{F}$ that fits both the train and the test sets (i.e., the discrepancy between $f(x)$ and corresponding $y$ is small for all $(x, y)$ in $(\mathcal{X}_{\text{train}}, \mathcal{Y}_{\text{train}})$

and $(\mathcal{X}_{\text{test}}, \mathcal{Y}_{\text{test}})$. However, when $\mathcal{X}_{\text{test}}$ is drawn from a different data distribution from that of $\mathcal{X}_{\text{train}}$, it has been widely reported that current deep learning models fail drastically [2, 8, 13, 32, 33]. This is because the models are inferred exclusively from the observed data distribution. Moreover, it could be the case that relations between $x$ and $y$ in testing samples are unseen during training, which raises questions on the feasibility of learning a single function when dealing with out-of-distribution tasks. A natural solution for this problem would be to train the model to learn the functions adaptively. Formally speaking, the model will learn a *function composer* $\phi\colon \mathcal{X} \times \mathcal{Y} \to \mathcal{F}$ that maps each pair of $(x, y) \in \mathcal{X} \times \mathcal{Y}$ (where $y$ is the associated output of $x$) to a function $\phi_{x,y}$ in $\mathcal{F}$ so that $\phi_{x,y}$ approximates the true relation between $x$ and $y$. As discussed, training models this way leads to two clear advantages: (1) it can help to handle the cases when there are multiple (and possibly disjoint) relations between inputs and outputs within the training and testing datasets; and (2) models are less dependent on data and thus can achieve more stable results on different training and testing sets. Empirical evidence for these points will be given in Section 4.

Since there are an infinite number of functions that can map an input to an output with any given degree of precision, the key is to design $\phi$ so that it can map a given input-output pair to a "good enough" function. We humans may draw analogies between the current situation and our experiences and then work out the most suitable options [12]. For example, we can recognize a math problem in exam to be similar to a previous exercise in class with different variable names. We may find the presented idea in a paper related to that in another paper we read before, as someone even says "new ideas are just re-distribution of old ideas". All these examples illustrate that analogy-making is a powerful strategy in human thinking process. Inspired by this mechanism, we equip the function composer $\phi$ with a semantic memory to store past knowledge, on which analogy-making is performed. The memory also plays the role of constraining the searching region for $\phi$, so that $\phi$ only looks for functions in the subspaces spanned by the memories instead of the whole functional space $\mathcal{F}$. Further analysis is given in Supplementary.

A remaining question is how to design the memory. Let us be inspired again by human thinking process: When we see an animal, we compare its face, legs or tail with things we know and finally conclude it's a "dog". Here "dog" is an abstract concept bound with primary characteristics (e.g. face, legs, tail, etc.); once a new entity arrives, we compare its properties with these characteristics, and if they are similar, we utilize the indirection mechanism to infer it is indeed the concept we are considering. In our case, the concepts are functions, or more specifically, the geometric transformations. We thus maintain a key-value memory structure, in which the keys represent abstract concepts and values the associated characteristics of the concepts. A new input-output pair matches with some values, and the indirection enables us to compute the functions based on corresponding keys.

Note that we have swapped the role of keys and values of the traditional memory (where the query is matched against the key, not value). This is to emphasize that we perform indirection to map concrete functional values to abstract functional keys.

## 3.2 Functional Indirection Neural Estimator

In this section, we present the Functional Indirection Neural Estimator (FINE), an neural architecture to realize the general idea laid out in Section 3.1. FINE learns to compose a function mapping an input embedding $x$ to an output embedding $y$ on-the-fly. The function is drawn from a parametric family specified by a backbone neural network. Coupled with the backbone is a *function composer* $\phi$, which is trained to compute the parameters of the backbone. More specifically, $\phi$ maps a data input-output pair to the weights of the neural networks.

FINE solves the proposed IQ tasks as follows: (1) Given two images $x, y$ from the hint, FINE uses $\phi$ to produce the function transforming $x$ to $y$; denoted by $\phi_{x,y}$. (2) Then, we feed the third image $x'$ to $\phi_{x,y}$ and get output $y^* = \phi_{x,y}(x')$. (3) We define a *similarity metric* to compare $y^*$ with given choices. The choice with the closest distance from $y^*$ is model's answer. The function composer and the similarity metric are specified as follows.

**Encoder**: Images are encoded by a trainable encoder (see Fig. 1). To effectively solve IQ tasks introduced in Section 2, we use the $p4$-CNN [7] encoder to serve as an inductive bias for geometric transformations. Throughout this paper, we refer to the images by their embeddings.

**The functional memory**

Let us focus a weight matrix $W_t \in \mathbb{R}^{d_t^{\text{in}} \times d_t^{\text{out}}}$ at the $t$-th layer of the backbone network. In practice, $W_t$ belongs to the huge $d_t^{\text{in}} \times d_t^{\text{out}}$-dimensional real matrix space $\mathcal{M}_{d_t^{\text{in}}, d_t^{\text{out}}}$. To reduce the complexity of $W_t$, we assume that $W_t$ only belongs to a $s$-dimensional subspace of $\mathcal{M}_{d_t^{\text{in}}, d_t^{\text{out}}}$, where $s \ll d_t^{\text{in}} d_t^{\text{out}}$. This subspace has a basis of $s$ matrices which are trainable and stored in FINE's memory, and $W_t$ will be written as a linear combination of these matrices.

Denote by $M_t$ the memory for the $t$-th backbone layer. $M_t$ includes two sub-memories: the key memory $M_t^{\text{key}} = \{M_{t,1}^{\text{key}}, \ldots, M_{t,s}^{\text{key}}\}$ and the corresponding value memory $M_t^{\text{value}} = \{M_{t,1}^{\text{value}}, \ldots, M_{t,s}^{\text{value}}\}$. Elements of $M_t^{\text{key}}$ and $M_t^{\text{value}}$ are trainable matrices of size $d_t^{\text{in}} \times d_t^{\text{out}}$. We further let $x_t \in \mathbb{R}^{d_t^{\text{in}}}$ and $y_t \in \mathbb{R}^{d_t^{\text{out}}}$ be the associated input and output, respectively, where $x_t$ is output of the $(t-1)$-th layer and $y_t$ is the pseudo-output computed by a trainable 1-layer neural network $y_t = \gamma_t(y)$.

With this design, we control the complexity of functional hypothesis space by either constraining the form of the backbone or the capacity of the functional memory.

**Memory reading**

By the virtue of simplicity, we aim to find a simple query that can demonstrate the relation between $x_t$ and $y_t$. Although the exact relation may be non-linear, we found that a query induced from linear relation is enough to efficiently read from memory. Formally, we want to find a query $W_t^q$ such that $W_t^q x_t = y_t$. The best-approximated solution is $W_t^q = y_t x_t^+$, where $x_t^+$ is the pseudo-inverse of $x_t$ and can be efficiently approximated by the iterative Ben-Israel and Cohen algorithm [3]. This way of query computing requires no parameter as opposed to other methods, where the input is often fed into a trainable neural network to compute the query.

With the query in hand, the next step is to perform analogy-making. In FINE, the query $W_t^q$ represents for the current situation and the value memory $M_t^{\text{value}}$ consists of past experiences. The concrete query interacts with value memories to measure how close the current situation is to each of the experiences. The similarities are computed as dot products between the query and value memories and normalized by a factor of $\sqrt{d_t^{\text{in}} d_t^{\text{out}}}$:

$$a_t = \frac{\text{fconcat}(M_t^{\text{value}})^\top \cdot \text{flatten}(W_t^q)}{\sqrt{d_t^{\text{in}} d_t^{\text{out}}}}, \tag{1}$$

where the flatten operator flattens the matrix $W_t^q$ of size $d_t^{\text{in}} \times d_t^{\text{out}}$ into a vector of size $d_t^{\text{in}} d_t^{\text{out}}$, while the fconcat operator first flattens all matrices in $M_t^{\text{value}}$, then concatenates them together to form the value matrix of size $(d_t^{\text{in}} d_t^{\text{out}}) \times s$. The resulting $a_t$ is a $s$-dimensional vector measuring the similarities between the query and the entries in the value memory. Here we omit the softmax operator as in usual attention to allow the similarities with more freedom. The same idea is shared in the ESBN [33], where the softmax similarities are scaled by a sigmoidal factor.

Finally, the value memories are bound with their associated key memories via indirection. This can be understood as moving forward from the concrete space of data and value memories to the abstract space of functions and key memories. With the key memories and the similarity vector $a_t$, the weight $W_t$ of current backbone layer can be computed as the linear combination of key memories:

$$W_t = \text{reshape}\left(\text{fconcat}(M_t^{\text{key}}) \cdot a_t\right), \tag{2}$$

where the reshape operator reshapes the vector of size $d_t^{\text{in}} d_t^{\text{out}}$ to a matrix of size $d_t^{\text{in}} \times d_t^{\text{out}}$. Since the softmax is omitted when calculating the similarities, $W_t$ is not constrained to be in the convex hull of the key memories and indeed can lie anywhere in the subspace spanned by those key memories.

**Memory update**

The key and value memories are updated using gradient descent:

$$M_{t,i}^{\text{key/value}} \leftarrow M_{t,i}^{\text{key/value}} - \lambda \frac{\partial L}{\partial M_{t,i}^{\text{key/value}}}, \quad \forall i = 1, 2, \ldots, s, \tag{3}$$

where $\lambda > 0$ is the learning rate and $L$ is the loss of the training step.

**The similarity metric**

After determining $\phi_{x,y}$, the model is given a new input $x'$ and being asked to select the correct associated output $y'$ among 4 choices $y'_1, y'_2, y'_3, y'_4$ so that $(x', y')$ follows the same transformation rule as $(x, y)$. This problem can be cast as finding the choice that is the most similar with $y^* = \phi_{x,y}(x')$. We consider the weighted Euclidean metric that measures the distance between two vectors $u, v \in \mathbb{R}^d$:

$$\eta(u, v) = \sum_{i=1}^{d} \alpha_i (u_i - v_i)^2,$$

where $\{\alpha_i\}_{i=1}^{d} \geq 0$ are trainable scalars, i.e., each component of $u$ and $v$ contributes with different importance. Finally, the probability to pick a choice is computed as:

$$p\left(y'_i \mid x'\right) = \frac{\exp(-\eta(y'_i, y^*))}{\sum_{j=1}^{4} \exp(-\eta(y'_j, y^*))}, \quad \text{for} \quad i = 1, 2, 3, 4.$$

## 4   Experiments

We conduct experiments to show the out-of-distribution generalization capability of FINE when performing tasks introduced in Section 2. For non-invertible transformations (e.g. reflection), we use a simple 2-layer MLP as the backbone. For invertible transformations, we use the NICE architecture [9] as backbone to serve as an inductive bias for invertibility. Since in each NICE layer, only half of the input is transformed, we use the same memory for two consecutive layers, i.e., $M_{2t} = M_{2t+1}$. To balance between the backbone complexity and computational cost, we use 4 NICE layers in all experiments.

We compare FINE with three major classes of models: (a) models that make analogies in the data space, including Transformer [30], PrediNet [29] and RelationNet [28]; (b) models that leverage indirection to bind feature vectors with associated symbols and reason on the symbols, including the ESBN [33]; and (c) models that aim to learn a mapping from data to the functional space, including the HyperNetworks [14]. For HyperNetworks, we still use the NICE backbone and just apply their fast-weight generation method for fair comparisons. Except for FINE and HyperNetworks, all models are trained with context normalization [32], which has been proved to be effective in improving the generalization ability.

**Datasets & implementation**: We generate data for IQ tasks described in Section 2 using images from Omniglot dataset [5], which includes 1,623 handwritten characters, and real-image CIFAR100 dataset [17]. If not specified, models are trained with $p4$-CNN encoder [7]. Experiments are conducted using PyTorch on a single GPU with Adam optimizer. Reported results are averages of 10 runs.

### 4.1   Results on Omniglot Dataset

| | Single-transformation | | | | | | | | | Multi affine |
| | Affine | | | | | Non-linear | | Syntactic | | |
| | Trans. | Rot. | Refl. | Shear | Scale | Fish. | H.Wave | B&W | Swap | |
|---|---|---|---|---|---|---|---|---|---|---|
| RelationNet | 27.1 | 26.2 | 25.5 | 27 | 27.5 | 26.1 | 30.2 | 49.7 | 26.0 | 25.3 |
| PrediNet | 68.5 | 43.9 | 32.9 | 62.4 | 65.7 | 36.2 | 46.1 | 60.5 | 57.5 | 34.9 |
| HyperNet | 88.9 | 62.0 | 94.0 | 74.5 | 81.8 | 63.2 | 80.4 | 88.6 | 90.1 | 54.0 |
| Transformer | 89.5 | 64.8 | 44.3 | 86.3 | 84.0 | 41.4 | 91.0 | 97.6 | 49.9 | 59.4 |
| ESBN | 79.8 | 58.6 | 50.1 | 84.5 | 83.4 | 67.1 | 86.4 | 90.5 | 71.6 | 63.1 |
| FINE-MLP | **96.1** | 73.9 | **95.1** | 84.5 | 86.0 | 70.4 | 84.8 | 94.9 | 87.5 | **72.0** |
| FINE-NICE | 94.3 | **77.6** | 57.7 | **87.2** | **86.6** | **78.5** | **95.9** | **98.4** | **96.2** | 69.1 |

Table 1: Test accuracy (%) on Omniglot dataset.

We use 100 characters for training and other 800 characters for testing. The train and test set size is 10,000 and 20,000, respectively. For FINE, we use 4 NICE layers with 48 memories for each pair

of NICE layers. Experimental results for single-transformation tasks are shown in Table 1. Overall, `FINE` dominates others with large margins. For example, the gap to the runner-up on the Rotation task is nearly 13%. With test accuracy over 75% on all tasks, `FINE` shows a strong capability of out-of-distribution generalization.

We further conduct multi-affine-transformation experiments. In this case, the training set includes multiple types of affine transformations, while other settings are similar to the single-transformation case. Results are also reported in Table 1. `FINE` continues to outperform other models. This is because only `FINE` explicitly assumes the existence of multiple good functions that represent the transformation from input to output data. We note that although HyperNet also makes a similar assumption by generating data-specific weights, it does not utilize analogy-making and indirection and thus, fails to generalize to unseen images.

## 4.2 Results on CIFAR100 Dataset

| | Affine | | | | | Non-linear | | Syntactic | |
|---|---|---|---|---|---|---|---|---|---|
| | Trans. | Rot. | Refl. | Shear | Scale | Fish. | H.Wave | B&W | Swap |
| RelationNet | 59.9 | 49.6 | 29.9 | 45.3 | 66.2 | 28.7 | 39.5 | 26.2 | 29.7 |
| PrediNet | 72.4 | 65.6 | 40.6 | 74.3 | 76.1 | 37.1 | 53.9 | 32.7 | 39.6 |
| HyperNet | 94.8 | 86.8 | 46.6 | 91.3 | 85.2 | 46.8 | 80.5 | 47.8 | 46.0 |
| Transformer | 98.4 | 86.3 | 47.5 | 95.4 | 84.9 | 47.2 | 95.1 | 51.6 | 47.6 |
| ESBN | 96.6 | 81.9 | 50.6 | 90.1 | 81.5 | 57.7 | 95.7 | 68.8 | 50.5 |
| FINE-MLP | 98.9 | 89.7 | **80.6** | **95.7** | 86.8 | 59.6 | 95.2 | 83.1 | 50.8 |
| FINE-NICE | **99.2** | **91.3** | 51.1 | 95.6 | **87.0** | **76.8** | **98.3** | **89.1** | **51.6** |

Table 2: Test accuracy (%) on CIFAR100 dataset of single-transformation tasks. For readability we report only the mean values here. Full table is reported in Supplementary.

| | RelationNet | PrediNet | Transformer | ESBN | FINE-MLP | FINE-NICE |
|---|---|---|---|---|---|---|
| Group CNN | $32.5 \pm 1.1$ | $46 \pm 1.0$ | $67.8 \pm 4.5$ | $71.1 \pm 0.5$ | $79.6 \pm 0.5$ | $\mathbf{81.6 \pm 0.5}$ |
| 3-layer ResNet | $31.1 \pm 2.7$ | $55.9 \pm 1.7$ | $28.5 \pm 0.5$ | $66.8 \pm 9.7$ | $\mathbf{77.8 \pm 0.5}$ | $73.5 \pm 1.6$ |
| MLP | $47.2 \pm 2.6$ | $59.3 \pm 0.8$ | $61.5 \pm 1.4$ | $54.7 \pm 2.6$ | $\mathbf{72.9 \pm 1.1}$ | $70.0 \pm 1.3$ |

Table 3: Test accuracy (%) on CIFAR100 dataset of multi-affine-transformation tasks.

For CIFAR100 dataset, we follow similar settings as in experiments on the Omniglot dataset, except that we use 50 classes for training and 50 remaining classes for testing. We also conduct experiments on single-transformation and multi-affine-transformation tasks. Results for single-transformation tasks are reported in Table 2. Again, `FINE` outperforms all other models on all tasks, especially on Reflection where the gap is nearly 30%. Although `FINE` does not show good performance on Swap task as in Omniglot experiments, it is still slightly better than other models. On the remaining tasks, `FINE` achieves test accuracy of more than 80%.

For the multi-affine-transformation task, we report the performances of the models when trained with different encoder architectures, including the $p4$-CNN, 3-layer ResNet and 2-layer MLP, in Table 3. The results show two superior characteristics of `FINE`: first, `FINE` is consistently better than other models across different encoders; second, `FINE` is more stable with small standard deviations. This empirical result supports the functional hypothesis stated in Section 3.1, where we suggest that focusing on functions distribution instead of data distribution can boost model's generalization capability and stability.

## 4.3 More Extreme OOD Tasks

Previous tasks only include unseen classes of objects during testing. In this section, we further test `FINE` and related models on more challenging OOD tasks: tasks with unseen rules during training and even ones with images from unseen datasets. The training and testing sets are either images from CIFAR100, Omniglot or MNIST datasets, while the hidden rules are either translation, rotation or shear. For translation, training problems consist of translation vectors $(a, b)$ with $|a|, |b| \leq 3$,

| Train set | CIFAR 100 | | | CIFAR100 | | | Omniglot | | |
| Test set | CIFAR 100 | | | Omniglot | | | MNIST | | |
| | Trans. | Rot. | Shear | Trans. | Rot. | Shear | Trans. | Rot. | Shear |
|---|---|---|---|---|---|---|---|---|---|
| RelationNet | 25.4 | 26.5 | 26.4 | 25 | 24.9 | 25 | 24.8 | 25 | 25.2 |
| PrediNet | 25.5 | 39.4 | 36.2 | 26.2 | 26.1 | 26.3 | 23.8 | 26.4 | 25.8 |
| HyperNet | 31.6 | 67.2 | 58.6 | 22.2 | 26.4 | 22.9 | 22.7 | 29 | 28.2 |
| Transformer | 30.9 | 64.2 | 55 | **30** | 31.7 | 33.8 | 28.4 | 27 | 25.1 |
| ESBN | 16.4 | 81.3 | 42.8 | 15.4 | 39.2 | 33.3 | 17.7 | 41.3 | 36.6 |
| FINE | **62** | **85.6** | **77.8** | 22.1 | **43.2** | **39.2** | **39.4** | **44.7** | **37.9** |

Table 4: Test accuracy (%) on more extreme OOD tasks with unseen objects, unseen rules and (possibly) unseen datasets. *Translation:* train with translation vectors $(a, b)$ with $|a|, |b| \leq 3$, test with either $|a| > 3$ or $|b| > 3$. *Rotation:* train with rotation angles $\alpha \leq 180°$, test with $\alpha > 180°$. *Shear:* train with shear angles $(\alpha, \beta)$ with $|\alpha|, |\beta| \leq 30°$, test with either $|\alpha| > 30°$ or $|\beta| > 30°$.

and models are tested with either $|a| > 3$ or $|b| > 3$; for rotation, model are trained with angles $\alpha \leq 180°$ and tested with $\alpha > 180°$; for shear, training angles $(\alpha, \beta)$ are ones with $|\alpha|, |\beta| \leq 30°$, while testing ones are either $|\alpha| > 30°$ or $|\beta| > 30°$. All tasks have 5,000 data points for training and 10,000 for testing. We report results of FINE with NICE backbone and related models in Table 4. As expected, FINE continues to outperform other models on most of the tasks, even on extreme OOD tasks with unseen datasets and unseen rules where performances of all models drop significantly. This demonstrates FINE is capable of effectively learning the basis weights, which are stored in the memory, to represent novel rules.

## 4.4 Model Analysis and Ablation Study

**Clustering on functional spaces**

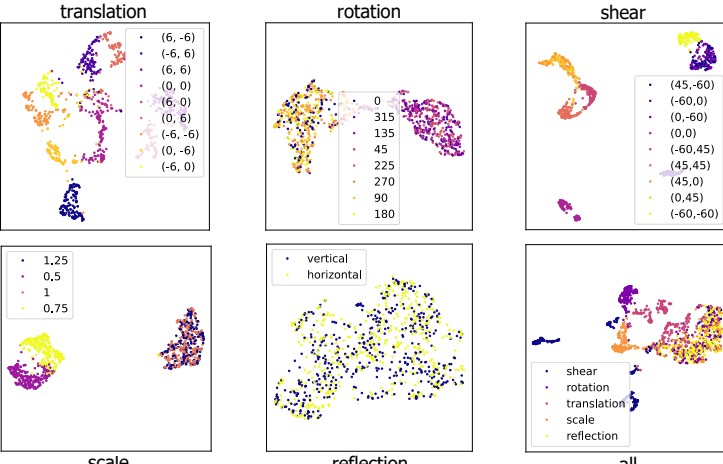

Figure 3: *Clusters on functional space.* We project weights produced by the function composer $\phi$ to see whether $\phi_{x_1, y_1}$ locates closely to $\phi_{x_2, y_2}$ on the functional space if the relation between $x_1$ and $y_1$ is similar to that between $x_2$ and $y_2$. Except for Reflection, the weights are nicely clustered.

We study how the transformations are distributed in the functional space. We use the FINE model trained on multi-affine transformation tasks. For an input-output pair $(x, y)$, we flatten and concatenate all weights of NICE layers to form a vector representing $\phi_{x,y}$. We then use the UMAP [21] to project $\phi_{x,y}$'s vectors onto the 2D plane. Results are shown in Fig. 3. It is interesting to see that shears with the same horizontal or vertical angles are positioned closely; and scales are separated into 2 "big" clusters, one for smaller scale and one for bigger scale. In contrast, reflection representations seem not to be clustered properly, which is worth further investigation in future work.

**Number of memories and backbone layers**

We do an ablation study to see the effect of number of memories and backbone layers in FINE. For the limit case with 0 memory, we assign the query matrices to be the weights for NICE layers without

the analogy-making and indirection process. For the limit case with 0 NICE layer, we replace the NICE backbone by a 2-layer MLP.

Results are shown in Fig. 4(a). Overall, we can observe clear improvements when we increase the number of memories or number of NICE layers. More interestingly, the more number of memories is, the more stable the results will be. Increasing the number of memories is equivalent to enlarging the range of $\phi$, and increasing the number of NICE layers is equivalent to enlarging the hypothesis space $\mathcal{F}$. Enlarging $\mathcal{F}$ may help FINE approach the true functions while still being sufficiently constrained by the number of memories, thus still being able to maintain its generalization capability.

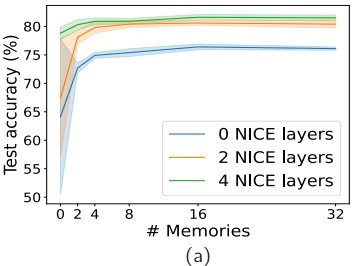
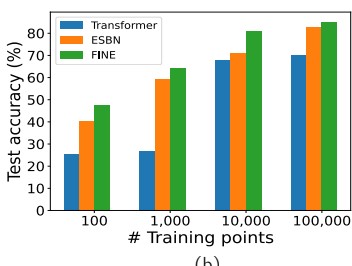

(a)  (b)

Figure 4: (a) *Performance of* FINE *with different number of memories and backbone layers*. Overall, the test accuracy increases with more memories and backbone layers. (b) *Performance with different number of training points*. FINE works fairly well on small datasets while others may fail significantly.

**Number of training data points**

We train models on multi-affine-transformation task with training sets of different sizes. Results are reported in Fig. 4(b). FINE can adapt with small datasets of sizes 100 or 1000 and achieve fair accuracy (47.8% and 64.4%, respectively). Moreover, FINE achieves average test accuracy of 81.1% on training set of size 10,000, which is quite close to the 85.3% accuracy on 100,000 data points training sets. This shows FINE may obtain a near-optimal solution even with small number of training data points. In contrast, ESBN and Transformer need 100,000 training data points to achieve 80% or higher test accuracy, while only achieving roughly 70% test accuracy on smaller datasets.

## 5   Related work

In recent years, there has been a strong interest in designing models that are capable of generalizing systematically. The Module Networks [1], which dynamically compose neural networks out of trainable modules, have been shown to possess some degree of systematic generalization [2]. Parascandolo et al. [24] proposed a method to train multiple competing experts to explain image transformations on MNIST and Omniglot datasets, yet the transformations are simpler than ones in our FINE dataset and it is not clear whether the proposed method can deal with unseen transformations. The Neural Interpreter [25] uses attention mechanism to recompose functional modules for each input-output pair and test their method on abstract reasoning tasks, yet tends to require a large amount of data to learn. Switch Transformer [10] mitigates the communication and computational cost in mixture of experts models. Recently, ESBN [33], which utilizes the indirection mechanism, shows a great promise on OOD tasks. Both methods achieve their degree of systematic generalization by injecting symbolic biases into the models. Our model FINE follows the same strategy, but it performs analogy-making and indirection in functional spaces instead of data spaces, and this has proved to boost both the performance and stability.

IQ tests are powerful testbeds for visual and abstract reasoning. Inspired by Raven's Progressive Matrices (RPM), the RAVEN dataset [34] was proposed as a testbed for visual reasoning models. However, this dataset does not focus on testing the ability of OOD generalization. Webb et al. [33] propose a series of IQ tasks with Unicode characters to show the effectiveness of indirection in tasks involving abstract rules, however these tasks are relatively simple since they only require the models to understand the same-different relation. The ARC dataset [6] aims to serve as a benchmark for general intelligence and includes various psychometric IQ tasks in the form of grid structures. In this paper, we propose IQ tasks involving geometric transformations as introduced in Section 2. These tasks are not only flexible so that we can include images from different datasets or create/combine numerous transformations, but also challenging to test OOD generalization abilities of models.

The weight composition feature of `FINE` links back to the concept of fast weights [19], the idea of computing data-specific network weights on-the-fly. HyperNetworks [14] stylizes this idea by computing the fast weights using a separate trainable (slow weight) network. The Meta-learned Neural Memory [23] uses the pseudo-target technique (which we also leverage in `FINE`) and appropriately updates the short-term memory once new input arrives. The Neural Stored-Program Memory (NSM) [18] proposes a hybrid approach between slow-weight and fast-weight to compute network weights on-the-fly based on slow-weight key and value memories. However, NSM only performs on sequential learning tasks, while `FINE` aims to solve OOD IQ tasks requiring abstract cognition. Moreover, `FINE` computes the query based on the input and the (pseudo-) output, while NSM's query is computed based on the input only. Memories have been to be versatile in meta-learning and few-shot learning [15, 27, 31] due to the ability to rapidly store past examples and adapt to new situations. In our case, an IQ task can be thought of as a one-shot learning scenario in which `FINE` has to make use of the long-term key-value memory to adapt to the current task.

## 6    Limitations

Operating in functional spaces requires `FINE`'s memory to store several trainable weight matrices to infer the data-specific weights for the backbone. Moreover, each layer of the backbone requires its own memory, thus `FINE` may need a large number of parameters for very deep backbones. This could be addressed by parameter-sharing across layers and limiting the rank of the weight matrices.

It remains to design the backbone architecture optimally. We have used the NICE architecture as backbone for invertible transformations and a 2-layer MLP backbone for non-invertible cases. We further investigated our model's performances with larger number of training points (up to 1M) and observed that `FINE` with the NICE backbone peaks at 100,000 training points with around $85\%$ test accuracy, while `FINE` with MLP backbone continues to improve and achieves around $95\%$ test accuracy at 1M training points. This calls for further theoretical analysis to guide the architectural design of the backbone network.

Finally, testing `FINE` on IQ tasks in the visual space may limit its potential on IQ tasks involving other modalities. For example, one may consider a textual IQ problem: "if abc $\to$ abd, then mmnnpp $\to$?". It is worth to emphasize that the concepts of analogy-making and indirection in functional spaces are indeed general and thus the idea of `FINE` should be applicable to various scenarios.

## 7    Conclusion

To study the out-of-distribution (OOD) generalization capability of models, we have proposed IQ tasks that require models to rapidly recognize the hidden rules of geometric transformations between a pair of images and transfer the rules to a new pair of different image classes. Such tasks would necessitate human-like abilities for conceptual abstraction, analogy-making, and utilizing indirection. We put forward a hypothesis that these mechanisms should be performed in the functional space instead of data space as in current deep learning models. To realize the hypothesis, we then proposed `FINE`, a memory-augmented neural architecture that learns to compose functions mapping an input to an output on-the-fly. The memory has two trainable components: the value sub-memory and the binding key sub-memory, where the keys are basis weight matrices that span the space of functions. For an IQ task, when given a hint in the form of an input-output pair, and `FINE` estimates the analogy between the pair and the values as mixing coefficients. These coefficients are then used to mix the binding keys via indirection to generate the weights of the backbone neural net which computes the intended function. For a test input of different class, the function is used to estimate the most compatible output, thus solving the IO task. Through an extensive suite of experiments using images from the Omniglot and CIFAR100 datasets to construct the IQ tasks, `FINE` is found to be reliable in figuring out the hidden relational pattern in each IQ task and thus is able to solve new tasks, even with unseen image classes. Importantly, `FINE` outperforms other models in all experiments, and can generalize well in small data regimes.

Future works will include making `FINE` robust against OOD in transformations without catastrophic forgetting when new transformations are continually introduced.

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
