# OpenReview forum: "Functional Indirection Neural Estimator for Better Out-of-distribution Generalization"
_NeurIPS.cc/2022/Conference — NeurIPS 2022 Accept_

### Official Review · Reviewer_JbkE · 2022-07-07

**Rating:** 5
**Confidence:** 4
**Soundness:** 2 fair
**Presentation:** 3 good
**Contribution:** 3 good

**Summary:**

This paper introduces FINE, a method for achieving out of distribution generalization through analogy making and indirection in the space of functions. FINE makes an analogy from a given input and output example to infer the function that ties them then uses indirection to approximate the function by composing a set of functions saved in memory. The paper introduces a visual reasoning dataset for evaluating out of distribution generalization. The dataset is based on standard vision benchmarks, CIFAR100 and Omniglot. Each sample consists of an input-output pair used as a cue for the transformation, an input image and 4 choices. FINE performs better than several comparable models over all functions and function combinations in terms of accuracy and sample efficiency. Ablation experiments highlight the importance of the number of layers and the memory size.

**Questions:**

These questions are directly related to the weaknesses mentioned above.
- The paper should clarify that the OOD generalization claim concerns the input images that undergo the transformations, not the transformations that are learned by the model. Evaluations on unseen transformations would be a valuable addition to the paper.
- The training setup for baseline models should be described in the supplementary material. Ideally, all models should be capable of inferring the transformation.
- As mentioned above, It would be a fair comparison to provide results on all models without NICE layers.
- Can this framework be adapted and evaluated on OOD generalization in image classification or other tasks in ML ?
- The first 2 paragraphs of section 3.1 could reformulated for clarification. As mentioned above, it would be more concise to write this section without using hypotheses. It’s simpler to explain FINE’s advantage when the task requires learning many functions with a single backbone.



**Limitations:**

The authors address certain limitations of the paper but address no negative social impact.

**Strengths And Weaknesses:**

Strengths:
- The paper takes inspiration from features of human intelligence and combines them in a novel way.
- It proposes a formal framework that uses this technique to mitigate the lack of OOD generalization in neural networks. It shows that the proposed method can abstract image-level transformations in the latent space and that the abstraction allows the model to generalize transformations to novel images.
- The paper proposes a novel use case in visual reasoning that highlights the new framework’s advantage. The method is compared to several relevant baselines.
- The paper is clearly written overall. The code is provided for reproducing the results.
- The authors discuss the certain limitations of their approach.

Weaknesses:
- To learn the proposed task, the model has to identify and build the transformation. Thus, the OOD samples should be different functions, not only different input images (for example, can the model learn rotations of angles between 0 and 90 degrees and extrapolate to angles of 90-180 degrees ?)
- To provide a fair comparison to baselines, all models need to be adapted to this framework. The training setup and hyperparameters used for training these models are not explained in the main paper or the supplementary material. This information is crucial for explaining their performance.
- The paper doesn’t discuss other applications for this framework. OOD generalization is an active research topic in ML and image classification is among the main applications (adversarial examples and domain shifts and noise corruptions are OOD examples). How can this framework benefit OOD generalization in this task?
- Although the use of NICE layers is intuitively motivated, it is not a necessary building block for FINE. Its use is motivated by the reversibility of certain transformations, which is a design choice in the dataset. Since most baselines (excluding hypernetwork) are not equipped with NICE layers, it would be a fair comparison to provide results on all models without NICE layers.
- The choice of using 2 NICE layers instead of one layer is not fully explained.
- The $y_t$ vectors used in the analogy step are output from $\gamma_t(y)$. The motivation and significance of this choice are not explained.
- In the first 2 paragraphs of subsection 3.1, H1 considers a unique output for each input while H2 considers the possibility of several outputs based on the transformation. The second hypothesis is not different from H1 if the target transformation is supplied with the input. This is the choice made indirectly in the dataset by providing an input-output pair as a hint for the transformation. It would be more concise to write this section without using hypotheses. It’s simpler to explain FINE’s advantage when the task requires learning many functions with a single backbone.

---

> ### Author Response · Authors · 2022-08-02
> **Response to Reviewer JbkE**
>
> We thank the reviewer for your detailed and insightful comments. We have updated our manuscript and would like to address the reviewer's concerns as follows:
>
> • “Evaluations on unseen transformations would be a valuable addition to the paper”: We thank the reviewer for the constructive suggestion. We agree with the reviewer and have added more extreme OOD tasks in the revision. In particular, the testing samples consist of unseen rules during training (as suggested by the reviewer), and we further construct testing problems with objects from unseen datasets (i.e. train and test on different datasets). Overall, our model continues to outperform competing models with an average margin of 7.9% to the runner-ups, which shows a clear superiority of our model on generalizing OOD. Please refer to Section 4.3 for more details.
>
> • “It would be a fair comparison to provide results on all models without NICE layers”: We have added experimental results of FINE with 2-layer MLP backbone in the revision. Overall, FINE with MLP backbone still performs better than competing models: on Omniglot dataset, FINE-MLP achieves the best performances on 2 single-transformation tasks, while maintaining second place (below FINE with NICE backbone) in the rest 7 single-transformation tasks; on CIFAR100, FINE-MLP achives the first and second place in 2 and 6 single-transformation tasks, respectively. This shows clear advantages of learning on functional spaces. Please refer to Section 4.1 and 4.2 for more details.
>
> • “The training setup for baseline models should be described in the supplementary material”: We have added the training setup in the Supplementary.
>
> • “The choice of using 2 NICE layers instead of one layer is not fully explained”: Since each NICE layer only changes half of the input, we need at least 2 NICE layers to completely transform the input. In the experiments, we use 4 NICE layers to balance the complexity of the backbone and computational cost. We have added this detail in Section 4.
>
> • “The vectors used in the analogy step are output from $\gamma_t(y)$. The motivation and significance of this choice are not explained.”: For an input-output pair $(x, y)$, we would like to estimate the weights of the neural network transforming $x$ to $y$, which means $x$ should the input for the first layer and $y$ should be the output of the last layer of the neural network. Hence, the outputs of intermediate layers are unknown. A common technique (e.g. see [1]) to estimate the intermediate outputs is to treat them as a function of $y$, i.e. $\gamma(y)$ where $\gamma$ is a trainable neural network.
>
> • “Can this framework be adapted and evaluated on OOD generalization in image classification or other tasks in ML ?”: The IQ problems introduced in our paper can be considered as an instance of one-shot learning in which the model is given a single hint. Therefore, we believe that FINE has the potential to successfully deal with few-shot learning tasks. However, for the clarity we did not include tasks other than ones introduced in the paper so that we can emphasize our 2 main contributions: the FINE dataset and FINE model. We believe applications of FINE on few-shot or meta-learning tasks are interesting and worth to investigate in future work.
>
> • “The first 2 paragraphs of section 3.1 could reformulated for clarification”: Thank you for the suggestion. We have revised the two paragraphs based on your thoughtful comment. In short, we clarify that learning a single function may be infeasible for OOD scenarios when there are new relations in the testing set, and that learning to compose functions adaptively may have a great potential on OOD tasks. Please refer to Section 3.1 for more details.
>
> To summarize, in the revision, we have added more extreme OOD tasks with unseen rules and unseen datasets, which we believe would be a valuable addition to our manuscript. We also rewrote Section 3.1 to improve clarification. We would like to emphasize that our paper makes 2 main contributions: first, we propose a new dataset for testing OOD generalization. Despite its simple appearance, our dataset remains challenging for most of the models, and moreover possesses a great flexibility to increase problem difficulties to fit with different OOD levels (e.g. unseen objects in the same dataset, unseen rules, unseen objects from unseen datasets). Second, we propose the FINE model that operates on functional spaces to learn to adaptively compose functions, and empirically show that FINE outperforms competing methods in various settings on our dataset.
>
> We hope that our responses have addressed the main concerns of the reviewer, and that the reviewer would increase their score accordingly.
>
> [1] Munkhdalai, Tsendsuren, et al. "Metalearned neural memory." Advances in Neural Information Processing Systems 32 (2019).

---

### Official Review · Reviewer_otSZ · 2022-07-09

**Rating:** 6
**Confidence:** 4
**Soundness:** 3 good
**Presentation:** 3 good
**Contribution:** 3 good

**Summary:**

This paper addresses the out-of-distribution generalization of deep learning models in IQ visual tasks involving extracting geometric transformation between a pair of images and applying the extracted transformation to a new image. It presents a memory-augmented neural architecture and on-the-fly model parameter retrieval from the memory to achieve OOD generalization in functional spaces.


**Questions:**

It's not clear whether the proposed method would work for other transformations.

**Limitations:**

Yes, the authors have addressed the limitations of their work.

**Strengths And Weaknesses:**

Strengths:
+ Unlike previous work that performs indirection and analogy-making in the data spaces, this paper proposes a mechanism for OOD in functional spaces.

+ The weights of backbone can be determined on-the-fly using (input, output) query to retrieve from a memory.

Weaknesses:
- The proposed method may need a large memory to store trainable weights for very deep backbones.

- The authors tested the proposed method only in functional spaces related to geometrical transformation. It's not clear whether the proposed method would work
for other transformations.

---

> ### Author Response · Authors · 2022-08-02
> **Response to Reviewer otSZ**
>
> We thank the reviewer for your comments. We have updated our manuscript and would like to address the reviewer's concerns as follows:
>
> • “It's not clear whether the proposed method would work for other transformations”: We included syntactic transformations in our experiments, including black-white and swap. These transformations are non-continuous and determined by pre-defined rules. Overall, our model performs better than competing models on the syntactic transformations.
>
> In the revision, we have added more extreme OOD tasks, including ones with unseen rules and unseen objects from unseen datasets (i.e. train and test on different datasets). Our model FINE continues to outperform others on most of the tasks with an average margin of 7.9% to the runner-ups, which shows a strong capability of FINE on generalizing on OOD samples. We believe this is a valuable addition to our paper.
>
> As conclusion, we would like to emphasize that our paper makes 2 main contributions: first, we propose a new dataset for testing OOD generalization. Despite its simple appearance, our dataset remains challenging for most of the models, and moreover possesses a great flexibility to increase problem difficulties to fit with different OOD levels (e.g. unseen objects in the same dataset, unseen rules, unseen objects from unseen datasets). Second, we propose the FINE model that operates on functional spaces to learn to adaptively compose functions, and empirically show that FINE outperforms competing methods in various settings on our dataset.
>
> We hope that the reviewer would satisfy with our responses and increase their score accordingly.

---

> > ### Comment · Reviewer_otSZ · 2022-08-06
> > **Raise Score**
> >
> > Thanks for your detailed response. The authors have addressed most of my concerns, so I would like to raise my score from 5 to 6.

---

### Official Review · Reviewer_kR5Y · 2022-07-10

**Rating:** 6
**Confidence:** 4
**Soundness:** 4 excellent
**Presentation:** 4 excellent
**Contribution:** 3 good

**Summary:**

This paper introduces a mechanism for functional indirection (FINE) in neural networks to achieve OOD generalization in abstract reasoning tasks. FINE proposes to dynamically select weights for a neural network backbone for a particular data input-output pair and use those weights to make prediction for an input that share similar hidden rule. The weights are selected from pre-defined key-value memory which is comprised of the weights that spans the space of possible functions. FINE is like a constrained form of hypernetworks where weights of the main (backbone) network are determined by another network. In this paper, this role is played by function composer $\phi$ which finds optimal weights and their arrangement for the main network using a limited basis of weights in the memory.

The paper further introduces a new abstract reasoning dataset based on Omniglot and CIFAR100 for evaluation.

**Questions:**

- Why create new set of IQ tasks when a similar benchmark of PGMs studying different forms of rule based OOD generalization exists already? It fulfils the criterion of providing hints of the hidden rule with few images and then using that to infer the predictions.
- Why are tasks restricted to identifying image transformations only? Would the method not be able to infer complex relational reasoning tasks as demonstrated in PGMs?
- How are the basis of network weights restricted such that it only spans to a limited set of functions that can be used by the backbone?
- Wouldn’t the limited basis of network weights restrict generalization to only observed rules and their combinations thereof? How would you scale to unseen rules?
- Some of the references in the related work section are missing for e.g. Neural Interpreters [3] uses attention mechanism to recompose functional modules for each input-output pair and test their method on abstract reasoning tasks. Similarly, Switch transformers[4] switch modules based on relevance to inputs.
- The model and the evaluation share similarities with meta-learning framework where a few input-output pairs are provided to adapt the network weights (e.g. in MAML) and the resulting model is used to make inference on the unknown inputs coming from the same classes. Analogous to the FINE datasets, the classes would be the hidden rule that input-output share. Curious to know what authors think about this and would it be possible to make a small experiment with Omniglot?
- Why is similarity metric chosen to identify correct output $y’$ instead of using normal cross-entropy?
- Do the newly introduced datasets test generalization only for unseen characters or also unseen rules? For e.g. the model could be trained for detecting character rotation from 0 to 90 degrees and tested on the characters rotated from 90 -270 degrees.

[3] https://arxiv.org/abs/2110.06399
[4] https://arxiv.org/abs/2101.03961

**Limitations:**

The main limitations seem to be around experimental evaluation. I would be happy to increase my score if authors address those concerns.


There doesn't seem to be any obvious negative societal impact.

**Strengths And Weaknesses:**

**Strength:**
- The proposed methodology of achieving functional indirection is novel and interesting.
- The paper is well written and easy to follow.
- The method is shown to achieve good performance on OOD generalization across unseen categories of the datasets used for training.


**Weaknesses:**
My main concerns revolve around the evaluation of method.
- The paper goal is to solve OOD abstract reasoning and analogy making. However, the method is not evaluated on the known abstract reasoning benchmark of PGMs. Instead authors propose similar but less complex tasks for evaluation.
-  The newly introduced datasets test generalization only for the unseen characters of the training dataset but not for unseen rules that could be extrapolated or interpolated. See me related comment in the questions section.
- The proposed method, FINE, implicitly selects the weights based on the transformation (hidden rule) of the input-output pair. This is highly similar to [1] where a mixture of expert (networks) explicitly compete to explain image transformations on MNIST and Omniglot. [1] also showed huge benefits of mechanism-specific function selection for OOD generalization. I believe a comparison or even explanation on the similarities among the method would be good to have in the paper.
- Since this framework focuses on the generalization of image transformation mechanisms across unseen classes (of Omniglot and CIFAR100), I would encourage authors to test FINE model trained with Omniglot transformations on MNIST data with transformation, which was shown in [1].

[1] https://arxiv.org/abs/1712.00961
[2] https://arxiv.org/abs/1807.04225

---

> ### Author Response · Authors · 2022-08-02
> **Response to Reviewer kR5Y**
>
> We appreciate the reviewer for your careful review and great pointers to related works. We have edited the manuscript accordingly and would like to address the reviewer's concerns as follows:
>
> • Questions regarding tasks with unseen rules or unseen datasets: In the revision, we have included more extreme OOD tasks, consisting of IQ problems with unseen rules or unseen objects from unseen datasets (i.e. train and test on different datasets). Please refer to Section 4.3 for detail. In short, FINE continues to outperform other competing methods with an average margin of 7.9% to the runner-ups in this extremely challenging OOD settings, thus further demonstrates the superiority of learning on functional spaces.
>
> • “Why create new set of IQ tasks when a similar benchmark of PGMs studying different forms of rule based OOD generalization exists already?”: There are 3 main reasons why we propose a new dataset for OOD problems. 1/ As far as we are aware of, the number of datasets for OOD tasks is still limited and there is no previous dataset that is as flexible as ours. Here OOD tasks are constructed at different levels: unseen objects from the same dataset, unseen rules, or even unseen objects from unseen datasets. Despite the simple appearances, our dataset remains challenging for previous models as demonstrated in empirical experiments. 2/ Previous PGM datasets, such as RAVEN [4] or PGM [5], may contain some critical issues that allow models to “cheat” on their problems, as reported in [6]. 3/ Recent inspiring works, such as [7], tested models on very simple IQ problems compared to ones in our FINE dataset. Along with the flexibility of FINE dataset as discussed above, the problem difficulties can be further increased, thus be able to adaptively fit with different testing goals.
>
> • “Why are tasks restricted to identifying image transformations only?”: For the scope of the paper, we only concentrate on geometric transformations to illustrate the idea of learning relations between objects. As far as we are aware of, real IQ tests involving images often require identifying image transformations. Moreover, using simple-looking image transformations allows us to easily generate data and control the task complexity, while still being able to increase problem difficulties whenever necessary. As shown in the empirical results in our paper, different types of image transformations (linear, non-linear, syntactic) along with images from various datasets and different training-testing strategies can help to construct from very easy tasks where all models can solve almost perfectly to very difficult tasks where models only randomly guess.
>
> • “How are the basis of network weights restricted such that it only spans to a limited set of functions that can be used by the backbone?”: The basis is constrained in the total number of weight matrices, and they are shared across all tasks, seen or unseen. The size of the support region of the matrices is therefore estimated by training across a variety of tasks.
>
> • Question regarding meta-learning: we agree with the reviewer that our proposed model is closely related with meta-learning framework. We did mention some related work regarding meta-learning and few-shot learning in Section 5. However, for the sake of paper completeness, we did not include applications of FINE on meta-learning or few-shot learning tasks as we would like to keep the readers focused on our 2 main contributions, which are the FINE dataset and the FINE model. We believe that meta-learning, few-shot learning tasks, or even OOD classification and regression tasks, are all interesting and potential applications of FINE, which we will investigate in future work.
>
> • “Why is similarity metric chosen to identify correct output instead of using normal cross-entropy?”: The predicted $y^*$ is a feature vector and not a probability distribution, thus we use the weighted MSE to compare the similarity between $y^*$ and $y^\prime_i$. Given the similarity, we compute the probability of selecting each image, and thus still use cross-entropy loss for learning.
>
> • Regarding the references [1], [2] and [3]: We thank the reviewer for their insightful references. We have added these references into the related work section as well as provided necessary analysis to compare with our model.
>
> To summarize, in the revision, we have added results on more extreme OOD tasks in Section 4.3 and relevant references in Section 5. We hope that our responses can address the main concerns of the reviewer, and the reviewer would increase their score accordingly.

---

> > ### Author Response · Authors · 2022-08-02
> > **Response to Reviewer kR5Y (cont.)**
> >
> > [1] Parascandolo, Giambattista, et al. "Learning independent causal mechanisms." International Conference on Machine Learning. PMLR, 2018.
> >
> > [2] Rahaman, Nasim, et al. "Dynamic inference with neural interpreters." Advances in Neural Information Processing Systems 34 (2021): 10985-10998.
> >
> > [3] Fedus, William, Barret Zoph, and Noam Shazeer. "Switch transformers: Scaling to trillion parameter models with simple and efficient sparsity." (2021).
> >
> > [4] Zhang, Chi, et al. "Raven: A dataset for relational and analogical visual reasoning." Proceedings of the IEEE/CVF Conference on Computer Vision and Pattern Recognition. 2019.
> >
> > [5] Barrett, David, et al. "Measuring abstract reasoning in neural networks." International conference on machine learning. PMLR, 2018.
> >
> > [6] Spratley, Steven, Krista Ehinger, and Tim Miller. "A closer look at generalisation in raven." European Conference on Computer Vision. Springer, Cham, 2020.
> >
> > [7] Webb, Taylor W., Ishan Sinha, and Jonathan D. Cohen. "Emergent symbols through binding in external memory." ICLR (2021).

---

### Official Review · Reviewer_WCRb · 2022-07-10

**Rating:** 3
**Confidence:** 5
**Soundness:** 2 fair
**Presentation:** 4 excellent
**Contribution:** 2 fair

**Summary:**

The problem of generalization to out-of-distribution (OOD) samples is key to most problems in AI. The authors address this challenge in the context of IQ-like tasks by introducing a method called FINE, for Functional indirection neural estimator. The authors are inspired by the idea of indirection, trying to connect two different representations and using one to learn or interpret the other. The authors show that the proposed architecture does well in those IQ tasks when considering images that are different from those in the training set upon applying the same rules as those in the training set.

**Questions:**

The paper starts with the assertion that humans can generalize to OOD. I am curious about the evidence for this. Sure, everybody says this sort of thing. But what kind of real quantitative evidence do the authors have for this statement?

On line 85, the authors assert that the models are required to recognize the objects and figure out the relation between them. What is the evidence for any of this? Sure, from introspection, humans may reason about the task in this way, but this does not mean that this is what the models are *required* to do. This is an example of anthropomorphizing.
(In contrast, lines 92/93 are trivially true. Yes, models must use the training data, there is no magic! )

The conclusions are similarly based on introspection only. The authors state that FINE reliably figures out the hidden relational pattern in each IQ task and is able to solve new tasks but the authors do not show any of this. The authors show that the model works better than a few other models when considering somewhat different images between the training set and the test set, which is pretty cool in and of itself. But there is nothing less and nothing more than that here. Other than somewhat different images, where does the paper show "new tasks" or "discovering relational pattern"?





**Limitations:**

The authors spell out some limitations mostly to highlight other interesting problems that could be addressed. The key challenges mentioned above are not discussed.

**Strengths And Weaknesses:**

Strengths

The question of generalization to OOD is fundamental to intelligence problems.

The idea of indirection is very interesting and the authors propose a very nice implementation of this idea.

Figure 4 is interesting in showing clusters in weight space. It would be interesting to show whether other models reveal the same property or whether this is specific to the proposed architecture.

Weaknesses

The term OOD is often used in a very loose manner. This study is a good example. To really define OOD, one should define D (i.e., the distribution). What I understand the authors are doing is selecting some “classes” in Omniglot and testing on other classes. Or selecting some “classes” in CIFAR and testing on other classes. This seems pretty standard. But the question is how much OOD is really being tested here. Imagine that your training class is letter “i” and your test class is letter “l”. Those letters are really similar. Sure, they are different “classes”. The issue is how similar the training set is to the test set. Can the authors train with Omniglot classes and test with CIFAR classes or vice versa? That would be impressive OOD!

The word generalization here only applies to testing with somewhat different images. The hard challenge in IQ tests is to generalize to novel rules.

---

> ### Author Response · Authors · 2022-08-02
> **Response to Reviewer WCRb**
>
> We thank the reviewer for your interests in our idea, implementation and findings, as well as for your insightful comments. We have revised our manuscript accordingly and would like to address the reviewer's concerns as follows:
>
> • Regarding quantitative evidence for human's capability to generalize OOD: To the best of our knowledge, there are some works providing quantitative evidence on human's ability to generalize OOD. The work in [1] measures performances of human and well-known deep neural networks (DNNs) on object recognition, and concludes that DNNs perform poorly when trained and tested on images with different distortion types, while human's performances are much more robust. If we switch our view to the field of neuroscience, we can further find more quantitative evidence, e.g. human can recognize visual scenes from line drawings and photographs with the same speed and accuracy [2], which shows a strong generalization ability on novel domains. We added relevant references to the revision to cover these quantitative evidence for human's ability on OOD generalization.
>
> • “On line 85, the authors assert that the models are required to recognize the objects and figure out the relation between them. What is the evidence for any of this?”: We thank the reviewer for pointing this out. We agree with the reviewer that our model takes inspiration from human's ability and that not every model is required to do so. We have rewritten related parts in the revision so that our approach is plausible and not a strict requirement to deal with OOD problems.
>
> • “The issue is how similar the training set is to the test set. Can the authors train with Omniglot classes and test with CIFAR classes or vice versa? That would be impressive OOD!”: We thank the reviewer for the nice suggestion. In the revision, we have conducted experiments that are “more” OOD: models are tested on problems with unseen rules and unseen images from unseen datasets. Overall, our model still outperforms competing methods with an average margin of 7.9% to the runner-ups. Please refer to Section 4.3 for detail.
>
> As conclusion, we would like to emphasize that our paper makes 2 main contributions. First, we introduce a new dataset to measure models' performances on OOD tasks. Despite its simple appearance, our dataset remains challenging for previous methods as demonstrated in experiments. Moreover, our dataset is flexible enough to offer OOD tasks with different levels: unseen objects from the same dataset, unseen rules, or even unseen objects from unseen dataset. Second, we propose FINE, which has been shown to work more effectively on different OOD scenarios induced from our dataset.
>
> We hope our responses can address the main concerns of the reviewer, and that the reviewer would increase their score accordingly.
>
> [1] Geirhos, Robert, et al. "Generalisation in humans and deep neural networks." Advances in neural information processing systems 31 (2018).
>
> [2] Biederman, Irving, and Ginny Ju. "Surface versus edge-based determinants of visual recognition." Cognitive psychology 20.1 (1988): 38-64.

---

### Meta-Review · Area_Chair_MbM9 · 2022-08-30

**Recommendation:** Accept
**Confidence:** Less certain

**Metareview:**

This paper tackles OOD generalisation through a mechanism for analogy-making in functional spaces rather than the data space. It involves construction of a functional framework that maps inputs to outputs---by abstracting the transformation between inputs and outputs through a separate (hyper)network which provides the weights of the mappings. It further contributes a benchmark for evaluating OOD on IQ tasks.

The reviewers agree that the paper tackles an interesting and relevant problem with the perspective of functional indirection, and the IQ task does appear challenging.

The primary outstanding issue with the work appears to be with what class of models they  compare against---there exists work in meta-learning (e.g. CAVIA, MAML) that ought to be discussed. If they are not appropriate it is crucial that this is explained, because from a functional perspective, they are very similar. Indeed as Reviewer kR5Y points out there are clearly relevant pieces of work that out to figure as comparisons here.
Simply speculating that 'it is not clear...can deal with unseen transformations' will not do; this ought to be established in order for the paper to stand strongly on its own.

And while I'm somewhat inclined to buy the proposition that PGM etc can have 'shortcuts' but I would perhaps have still expected evaluations on these in addition to the IQ task setup to mitigate claimed deficiencies in benchmarks like RAVEN or PGM.

The authors also provided additional experiments over more extreme OOD settings as well as tempered some imprecise statements to address reviewer concerns, which was good.

On balance, though it appears as if the paper has more merits than issues, and most of the issues raised could be addressed with some work. I would strongly urge the authors to actually make the edits for comparison and incorporate the additional experiments over existing benchmarks from the rebuttal into the manuscript, as requested by the reviewers.

**Award:**

No

---

### Decision · Program_Chairs · 2022-09-14

Accept